# Modification of Hot-Melt Adhesives Based on Metallocene Poly(ethylene-propylene) Copolymer for High Adhesion to Polar Surfaces

**DOI:** 10.3390/polym14061253

**Published:** 2022-03-20

**Authors:** Igor Novák, Jozef Preto, Vladimír Vanko, Jozef Rychlý, Juraj Pavlinec, Ivan Chodák

**Affiliations:** 1Polymer Institute of the Slovak Academy of Sciences, 845 41 Bratislava, Slovakia; upolnovi@savba.sk (I.N.); upoljory@savba.sk (J.R.); upolpav@savba.sk (J.P.); 2VIPO, 958 01 Partizánske, Slovakia; jpreto@vipo.sk (J.P.); vvanko@vipo.sk (V.V.)

**Keywords:** hot-melt adhesives, metallocene polyolefins, adhesion strength, acrylic acid grafting

## Abstract

A procedure is described of grafting the acrylic acid onto an oxygen/ozone-activated metallocene poly(ethylene-co-propylene). Consequently, the grafted copolymer is applied as a component in a metallocene polyolefin-based hot-melt adhesive composition with increased adhesion. The surface properties and adhesion strength of the prepared hot-melt adhesive (HMA) were determined and used to account for the effect of grafting. The application of grafted polyolefin as one of the components of the HMA mixture provides significant increase in adhesive strength, and it also results in increased compatibility and negligible effects on the technological parameters of the final composition. The obtained results may have significant impact for the practical application of prepared HMA for book bonding.

## 1. Introduction

Polyolefins are often used as a major component of hot-melt adhesives (HMAs), containing in many cases a rather complex composition of waxes and tackifiers to achieve the desired rheological and adhesive properties. The final composition of a particular HMA must be carefully designed so that all processing parameters and application properties are optimal for the intended application. Even small modifications of the designed composition may lead to unexpected problems either during processing or later in the application phase. Thus, when designing thermoplastics-based HMAs for bonding more polar materials (such as paper, cardboard or wood), it is useful to modify the nonpolar chains of the polymers used with polar monomers to achieve higher-polarity HMAs to ensure stronger adhesive joints. In addition to the polarity of the modified HMA, the rheology of the polymer melt is also of key importance, because, for the appropriate bonding of paper, cardboard or wood, mechanical adhesion is ensured by anchoring the HMA in the pores on the material surface [1].

Metallocene polyolefins (MePOs) are highly hydrophobic polymers with low surface energy and especially small polar component. Recently, several papers have been published describing the application of metallocene polyolefins as components that provide excellent performance to HMAs [1,2]. Therefore, for the preparation of HMA-containing MePO [3,4,5] for bonding more polar substrates, it is desirable to increase the polarity of the adhesive either by adding a polar low-molecular weight or polymeric additive or to perform chemical modification of the MePO by incorporating additional polar moieties into the HMA polymer blends. A modification of one of the polyolefin species present in the HMA composition by polar functional groups of selected polar monomers to the nonpolar MePO chains may be preferred since the increase in the hydrophobicity/polarity does not substantially alter the other key properties, especially the mechanical properties and rheological parameters of the final HMA [1].

The grafting of the selected monomer onto the MePO polymer backbone takes place by a free-radical mechanism. The formation of hydroperoxides and peroxides on polymer chains by oxidation may be initiated by thermal decomposition of organic peroxides (e.g., benzoyl peroxide, or dicumyl peroxide) or by radiation [6,7,8,9,10].

To increase the polarity, it is advantageous to modify MePO in the form of a powder and to inoculate polar vinyl monomers in the presence of a free-radical initiator. Favorable conditions with higher cleavage efficiency exist for the metallocene poly(ethylene-co-propylene) copolymers containing hydrogen on tertiary carbon, which are more reactive in transfer reactions [11]. The introduction of the species potentially initiating free-radical grafting may be easily achieved by ozone [12,13,14,15]. The advantage of such a procedure consists in the fact that both initiator and grafts are attached to the same polymer chain [16,17,18,19,20], so that the expected increase in the polarity of the composition does not need any introduction of an additional additive, which might affect the processing parameters and/or ultimate properties of the original material. The formation of peroxides on polymer chains can be significantly accelerated by using a mixture of ozone and oxygen.

However, a number of different schemes has been published in the literature related to the reaction of ozone with metallocene polymers (in most cases, the substrates are other polyolefins or completely different polymers) and none of these are generally accepted. Since this contribution is aimed to ultimate properties of HMAs, we are not going to discuss that aspect in detail; however, in our view, among several alternative reaction schemes, the most acceptable one seems to be the ozone interaction with double bonds present in small concentrations in all polyolefins and further formed during the early stage of the ozonation. The process results in Criegee’s molozonides [21] followed by subsequent cleavage and formation of various species including carbonyl hydroperoxides, as shown in Figure 1.

Further scission of these hydroperoxides may lead to grafting as well as to reduction of the polymer chain length [22].

The formation of peroxides on polymer chains can be significantly accelerated by using a mixture of ozone and oxygen. After the grafting of more polar chains in the form of side branches on the main polymer chain, the surface energy and in particular its polar component, are increased, while the difference between the polarities of HMA and the bonded polar substrate is reduced, resulting in high adhesion [23]. The grafted MePO will then be used to formulate new HMA with increased adhesion.

However, the addition of even one new component to the optimized composition of the HMA may lead to a destruction of the delicate balance of a full set of key ultimate properties and processing parameters. Therefore, it seems to be benefiting to perform any changes through modification of a small part of the component that is present in the original mixture. By such a way, the compatibility between the polymers, as well as e.g. viscosity or thermal stability, may be kept close to the initial optimal values, making the optimization of the composition much easier. From this point of view, the utilization of gaseous ozone appears to be beneficial.

The presented paper investigates the procedure of the grafting of acrylic acid onto an oxygen/ozone-activated metallocene poly(ethylene-co-propylene) copolymer. The grafted MePO is consequently applied as a component of the HMA to increase the adhesion of the MePO-based HMA composition. Thus, the final HMA recipe contains the same MePO for both the original and acrylic acid-grafted samples. The probability of substantial changes in key parameters and crucial properties of the modified HMA would be minimalized or eliminated by such a way. The surface properties and adhesion of this system to model paper materials were determined and used to account for the effect of grafting.

## 2. Materials and Methods

### 2.1. Materials

The random metallocene ethylene-co-polypropylene copolymer Licocene PP 2602 (Clariant, Muttenz, Switzerland), with T_m_ = 95,102 °C, T_g_ = 57 °C, density = 0.874 g/cm^3^, tensile stress at yield = 1.76 MPa, was used as the main component of the HMA, as well as for the preparation of the grafted component modified by ozonation and grafting by acrylic acid.

Other polymers included in the overall HMA composition were Licocene PP 3602 (a low crystalline metallocene propylene–ethylene copolymer, Clariant) and PP 1302 (a metallocene propylene–ethylene co-polymer wax, Clariant). Licowax PE 520—a medium-molecular weight nonpolar polyethylene hard wax (Clariant) with dropping temperature T = 120 °C was used for viscosity adjustment, and Escorez 5600 (Exxon Mobil Chemical, Houston, TX, USA)—an aromatic modified, cycloaliphatic hydrocarbon resin—was applied as tackifier.

Acrylic acid (AA) (99% monomer purity, Sigma-Aldrich, St. Louis, MO, USA), stabilized with 180–200 ppm 4-methoxyphenol, was used for grafting, while sodium laurylsulfate (NaLS) (purity ≥98%, Aldrich), was added as a wetting agent in the grafting process. Kinox-10—pentaerythrityl tetrakis [3-(3,5-di-tert.butyl-4-hydroxyphenyl) propionate] (HPL Additives, Haryana, India)—was used as an antioxidant. Regalite R 1100—a low molecular weight fully hydrogenated hydrocarbon—was used as a tackifying resin (Eastman, Kingsport, TN, USA).

### 2.2. Acrylic Acid Grafting on Licocene PP 2602

The modification of Licocene PP 2602 by grafting the polymer in powder form with a polar monomer was performed by activation of the polymer molecules with a mixture of O_2_ + O_3_ as the first step. Ozone was generated by passing oxygen through a Profizon X ozone generator (UVC Servis, Prague, Czech Republic), enabling excellent control of hydroperoxide (HPx) and peroxide (Px) formation. The ozone concentration in the mixture with O_2_ was 17.8 mg/L.

The absolute concentration of HPx and Px groups bonded to LC was determined by modified iodine analytical procedure (16). Moreover, the presence of peroxides, their thermal stability and kinetics of decomposition were monitored in parallel by chemiluminescence (CL) using a LUMIPOL 3 photon-counting instrument (commercial product of Polymer Institute SAS, Bratislava, Slovakia). The measurements proceeded under non-isothermal conditions in a temperature range between 50 and 250 °C, with a heating rate of 5 °C per minute. Thermal heat emission under the given conditions is negligible compared to the CHL intensity from sample degradation. The instrument dark count rate was 2–3 counts/second, and the resolution level at 40 °C was 2 photons/second.

Modification of ozonated Licocene PP 2602 (LC) with acrylic acid proceeded in a 50 mL chamber of a Brabender Plasticorder PLE 331 laboratory internal mixer (Brabender, GmbH, Duisburg, Germany). Considering the polymer melting temperature of 95–102 °C and thermal stability of hydroperoxides present in ozone—modified LC as determined by chemiluminescence, the grafting temperature was set to 120 °C and the intensity of mixing was 30 rounds per minute (rpm). No special arrangement was applied to maintain the oxygen-free atmosphere. The optimized composition of the reaction mixture consisted of basic LC polymer mixed with 18.5 wt.% of acrylic acid, while 0.5 wt.% of NaLS wetting agent was added to obtain increased homogeneity of polar acrylic acid in apolar metallocene polyolefin. After completing the grafting reaction, the content of acrylic acid in the final product was 11.4 wt.% according to FTIR analysis.

The procedure of grafting proceeded as follows: the temperature of the reaction mixture and the torque in the Brabender chamber depending on the mixing time show that the polymerization reaction of the grafting begins shortly after the monomer is added to the system. The temperature set in the mixing chamber was 120 °C and was monitored by a bimetal thermometer with its top directly in the mixing chamber with intimate contact with the polymer melt. After filling the chamber with the polymer and closing the chamber, the temperature decreased to about 80 °C within 5 to 10 s, and monomer acrylic acid was added within a further 10 s. Maximum temperature in the chamber of 129 ± 0.8 °C was achieved after 11.5 min as a result of both intensive mechanical mixing and exothermal chemical reaction. During the initial heating and adding liquid monomer, the torque and the melt viscosity decreased, but with the onset of polymerization as the amount of monomer reacts and the grafted copolymer is formed, both the torque and the viscosity of the system increased. Rapid grafting reaction proceeded between 10 and 15 min with a maximum at 11.5 min, indicated by an increase in the torque. The reaction was over after 22 min.

### 2.3. Experimental HMA Preparation and Characterization

The grafted Licocene copolymer prepared according to the previous section was used in varying amounts as the additive for the experimental HMA. The composition of the basic material is shown in Table 1. The variations in composition given in Table 1 for materials HMA prepared to estimate the effect of other components are described in the legends of the corresponding Tables or Figures.

The samples for analytical procedures were prepared from the LC-grafted product by compression molding to disks 1 mm thick with a 20 mm diameter at a temperature of 180 °C, the molding time was 180 s, and the specific plate pressure was 3 N/mm^2^.

Brookfield viscosity was measured according to STN EN 2555. The softening temperature was measured according to STN EN 1427. Solidification time determination was performed according to the VIPO internal standard by means of a prototype equipment for the examination of glue joints developed in ZDA Partizánske, Slovakia. In this procedure the bonding material was uncoated paper with a basic weight of 500 g/m^2^. The required testing temperature was adjusted in the equipment and paper strips with dimensions of 6 cm × 1.5 cm were placed into the equipment. The expected time of solidification was set up on the timer. One bead of tested molten hot melt was applied on the bottom paper strip, and the upper paper strip was immediately placed on the top. In this way a glue joint with measurement area of 1.5 cm × 1.5 cm was created and the timer was switched on. When the time was over, a weight of 2 kg was applied in shear mode on the glue joint. If the glue joint failed, the time interval in the timer was increased and the test was repeated step by step until the glue joint held up the weight. This time interval was defined as solidification time.

Open time was measured according to the VIPO internal standard. A hot-melt film with a width of 0.5 mm was prepared using a hot-melt spreader. The hot-melt film was placed on a nonabsorbent plate and put in an oven adjusted to the required temperature. After 10 min, during which the hot melt became molten, the plate with the hot melt was placed on an insulated plate and the paper strips from the tested paper with an area of 20 mm × 60 mm were placed one by one on the hot melt and pressed by a 200 g weight in order to obtain the 20 mm × 20 mm bonded surface. The procedure was repeated, keeping the selected time intervals between the individual steps of the paper piece bonding test. After 2 h of conditioning at room temperature, the paper strips were pulled out by hand. The open time was determined as the maximum time interval when the glue joint still failed in the cohesive tearing of paper.

### 2.4. Surface Energy Measurements

The polarity/hydrophilicity of metallocene polyolefin increased after grafting of the polymer with AA. Based on static contact angle (CA) measurements of a set of three testing liquids [18], namely redistilled water, glycerol and dimethyl sulfoxide, the free surface energies were determined. The volume of a drop of the testing liquid was 3 μL. Ten separate values of each CA were averaged to obtain one representative contact angle value for each liquid. The hydrophilicity of LC-g-AA surfaces was evaluated and the values of the surface energies were determined. A Professional Surface Energy Evaluation (SEE) system with a CCD camera (Advex Instruments, Brno, Czech Republic) was used for experiments and the sessile drop technique was applied. The contact angle of each drop was measured approximately 3 s after the drop was placed, which was sufficient for achieving thermodynamic equilibrium between the solid, liquid, and gas phases. The data were used for determination of the total (*γ_s_**^tot^*), polar (*γ**_s_**^p^*) and dispersive (*γ**_s_**^d^*) components of the surface free energy. The surface energies of the polymer were evaluated by the Owens–Wendt–Rable–Kaelble (OWRK) method [18,19]:(1)$$\frac{\left(1+\cos\theta \right)\gamma_{LV}}{2}=\left(\gamma_{LV}^{d}\gamma_{s}^{d}\right){\;}^{1/2}+\left(\gamma_{LV}^{p}\gamma_{s}^{p}\right){\;}^{1/2}$$
where
*θ* = contact angle of testing liquid (deg),*γ_LV_* = surface free energy (SFE) of the testing liquid (mJ·m^−2^),*γ_LV_^d^, γ_LV_^p^* = dispersion component (DC), and polar component (PC) of the testing liquid SFE (mJ·m^−2^),*γ_s_^d^, γ_s_^p^* = DC and PC of the polymer SFE (mJ·m^−2^),*x_s_^p^* = *γ_s_^p^/γ_s_^p^ + γ_s_^d^* = polar fraction.

### 2.5. Adhesive Properties

To assess the strength of adhesion of paper to the HMA based on the modified MePO, a specific method was developed. HMA samples were tested to verify the reproducibility. The adhesive joints were prepared by applying a 500 µm thick film of adhesive on a medium-density fiber board (MDF) and consequently anchoring 50 sheets of 135 g/m^2^ paper with an area of 90 mm × 90 mm on the glued side of the board. Individual sheets/pages were subsequently ripped off in the specified order (page order) using an Instron 4301 universal testing machine (Instron, Norwood, MA, USA) equipped with a 5 kN measuring cell as shown in Figure 1. The adhesion value was determined as the tensile strength, calculated in N per 90 mm, of the adhesive joint of HMA paper to pull out one paper sheet. Measurements of the strength of adhesive joints were tested at 9 places of the testing paper binding (on pages 6, 10, 15, 20, 25, 30, 35, 40 and 45) and the final extent of adhesion was expressed as average values of adhesive strengths and statistical deviations.

## 3. Results and Discussion

### 3.1. Formation of Peroxide and Hydroperoxide Moieties by Ozonation

The concentration of hydroperoxides formed on Licocene PP 2602 after ozonation depends on the temperature, ozone concentration and time of reaction. In our case, at a constant temperature and O_3_ content in the reactive gas, the desired concentration of peroxide and hydroperoxide groups attached to the polymeric macromolecules is easily controlled by changing the ozonation time, as seen in Table 2. The increase in HPx moieties with time is close to linear.

Since thermal stabilities and decomposition kinetics are important characteristics of HPx, the decomposition of HPx in the activated polymer was controlled by temperature and monitored by titration as well as by the CL method under nitrogen. The oxidized polymers also emit photons when heated in an inert atmosphere to a temperature high enough to decompose peroxides.

The dependence shown in Figure 2 confirms that several types of HPx groups with different thermal stability are formed on the macromolecule backbones of LC during ozonation. The least stable HPx with measurable decomposition from 50 °C, and maximum at 90 °C and 98 °C is formed predominantly at the beginning of ozonation. The concentration of this type of HP_x_ gradually decreases and the subsequently formed HPx moieties are thermally more stable. The primary HP_x_s observed at the onset of ozonation appear to be transformed into more stable HPx types over the continuous course of ozonation. Interestingly, the HP_x_ moieties formed on the metallocene polypropylene–ethylene copolymer molecules decompose in the temperature range of approximately 80–120 °C with a maximum at 105 °C, while more heat-resistant HPx are also present with a maximum decomposition temperature at 144–146 °C.

The FTIR analysis of pristine LC- and AA-grafted LC (plot b) is illustrated in Figure 3. The presence of grafted PAA in LC-g-AA was proven by FTIR analysis. The bands of functional groups C-O and COC (1715 cm^−1^ and 1170 cm^−1^, respectively) indicate the presence of PAA chains. The ratio of the FTIR absorbance intensities of acid carbonyl compounds at 1715 cm^−1^ and the reference band at 1464 cm^−1^ was used to determine the PAA concentration in the grafted copolymer.

### 3.2. Properties of Mixtures Containing Grafted Licocene 2602

The results of the surface property measurements of PAA-grafted LC are summarized in Table 3. A decrease in the water contact angles was observed with an increase in the concentration of LC grafted with 11.4 wt.% of PAA from 95.9^O^ for reference virgin nongrafted LC to 67.4° (27.3 wt.% for LC grafted with 11.4% of PAA), and the polar component of the SFE increased four times from 1.2 to 4.8 mJ·m^−2^. On the other hand, grafting AA onto LC did not lead to any change in the dispersive components. The polarity of the polymers as well as the HMA expressed by a polar fraction is represented by the ratio of the polar component of the free surface energy to the total surface energy, which is given by the sum of the free surface energy of the polar and dispersion component. The polar fraction (*x_s_^p^*) of LC during grafting significantly increased from 0.04 (pristine sample) to 0.14 (sample with 27.3 wt.% of PAA).

The impact of an aromatic-aliphatic resin-based tackifier Escorez 5600 was also tested to determine its impact on the strength of the adhesive bond HMA paper. The tested composition contained 79.5 wt.% LC, 0.5 wt.% Kinox 10 antioxidant, and various amounts of the tested tackifier resin. The values of the resulting strength of the adhesive joints with the tackifier resin Escorez 5600 are shown in Figure 4. The effect of the tackifier consists in a decrease of viscosity and the increase of wettability; both effects contribute to the adhesion strength increase. On the other hand, the cohesive strength of the tackifier is lower, compared to the polymeric components of the HMA. The superposition of the two counteracting effects may be responsible for maximal adhesive strength for HMA containing 20 wt.% of the tackifier.

To find an optimal composition of the HMA suitable for the bonding of book backs, the impact of the wax Licowax PE 520 on the properties of the HMA was investigated. The effect of the wax addition to the composition on the HMA adhesion strength is shown in Figure 5. Licowax PE 520 was chosen due to its suitable open time, setting time, and good compatibility with LC, which is demonstrated by the high elasticity of the adhesive film. However, as seen in Figure 5, the presence of Licowax PE 520 leads to a significant reduction in the tensile strength of the adhesive joint of HMA paper. The tensile strength of the adhesive joint of HMA paper, as shown in Figure 4, decreased significantly from 163.4 N/90 mm (0 wt.% wax, A) to 71.7 N/90 mm (5 wt.% wax, B) and to 51.4 N/90 mm (8 wt.% wax, C). The reason may be seen in the decrease in the cohesive strength of the HMA due to the addition of a low-molecular-weight additive.

### 3.3. Properties of the Hot-Melt Adhesive with Increased Adhesion

After testing several individual polymeric additives concerning the effect on the adhesive strength of the HMA, the final composition was prepared according to Table 1. The properties of this material are shown in Table 4, along with the results of surface property measurements, i.e., contact angles of selected testing liquids, surface free energy, including its polar and dispersion components. The water contact angle (WCA) decreased with increasing polar component in the HMA (from 96.8 to 72.8°), and the polar component of the free surface energy increased from 1.8 to 3.3 mJ·m^−2^.

Figure 6 summarizes the effect of the polar additive on the strength of the adhesive joint of HMA paper for various concentrations of added polar additive (L3 in Table 1) in the HMA. The effect of polar content on the HMA adhesion strength is statistically significant and technologically interesting for the sample with the highest content, 0.544 wt.% of the polar additive (Figure 6), which reached a strength value of the adhesive joint of 142.3 N/90 mm compared with an adhesion of 117.9 N/90 mm for the pristine sample, justifying the application of the AA-grafted polar modifier.

## 4. Conclusions

Ozonation of one of the main polymeric components of the hot-melt adhesive, namely poly(ethylene–co-propylene) Licocene PP 2602, leads to the formation of peroxides attached to the polymer chain which, in a subsequent decomposition, initiate the grafting of highly polar acrylic acid on it. The addition of a small amount (0.544 wt.%) of such polymeric additive grafted with highly polar functional groups results in a significant increase in the adhesion of HMA to polar surfaces via an increased surface free energy of the HMA. This appears to be beneficial compared to introducing other additives that might be incompatible with the HMA components. Thus, the substantial changes of the processing parameters and the ultimate properties of the HMA, such as melt viscosity, solidification time and softening temperature, are reduced.

## Data Availability

No additional data are publicly accessible.

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
