# Peer review of "Modification of Hot-Melt Adhesives Based on Metallocene Poly(ethylene-propylene) Copolymer for High Adhesion to Polar Surfaces"

_polymers, 2022, doi:10.3390/polym14061253_

Round 1

Reviewer 1 Report

In my general opinion, the manuscript is written carelessly. As a reader, understanding the entire research process created a huge problem for me. I am not fluent in English, but I believe the article should be revised by nativspeker.

In my opinion, optimization has not been performed in the work - as the title suggests. No models describing the change of the properties of hot-melt adhesives due to variable components of the mixture have been determined.

In my opinion, optimization has not been performed in the work - as the title suggests. No models describing the change of the properties of hot-melt adhesives due to variable components of the mixture have been determined.Optimization is a method of determining the best (optimal) solution (searching for the extreme of a function) from the point of view of a specific criterion (indicator). Testing whether a specific additive reduces or increases a certain property of a material is not an optimization. Additionally, the study did not investigate the interactions related to the simultaneous presence of several components in the mixture. The authors of the paper often define the changes as significant, although the paper does not present even the simplest statistical analysis determining the significance of the observed changes. After such an analysis, it might turn out that some modifications do not affect the properties of the obtained glued joints.

The research methodology should be clarified. The method of activating the polymer with O3 (time, temperature, polymer form, apparatus) has not been sufficiently described. The process of preparing the mixture should be more explained. For example: at what point and what ingredients were added to the mixer. The description shows that only Licocene PP 2602 was mixed with the monomer - at which point the remaining ingredients were added? How was Escorez 5600 or Licowax introduced? How was the sample preheated to 80 °C when the temperature of 120 °C was left on the chamber?

Table 1 does not include all analyzed systems.

The description of the research results is also difficult to understand. The process of preparing the mixture is imprecise and general. It does not show at what point which component was used - heating 0-5 s? suitably the 5-10 S monomer? - monomer was added gradually. No explanation of the increase in torque and temperature of the mixture after 5 minutes (maybe it is due to my misunderstanding resulting from the lack of diligence in the editing of the text).

There is no analysis of the presented results of rheological tests. Rheological studies are incomplete and do not take into account all the tested systems.

There are really many such uncertainties in the manuscript and I will not mention them anymore.

The list of literature has been made carelessly and there are no recent reports.

In my opinion, the article should be thoroughly revised. In my opinion, the scientific level of the presented research is inadequate to the rank of the Polymers journal

Author Response

REVIEWER 1

In my general opinion, the manuscript is written carelessly. As a reader, understanding the entire research process created a huge problem for me. I am not fluent in English, but I believe the article should be revised by nativspeker.

I apologize for not making clear that the detailed English corrections were done before submitting the first version of the MS to the journal. The English corrections were made by American Journal Experts, THE CERTIFICATE IS ATTACHED. The review was done before submitting the primary version of the manuscript to the POLYMERS (pls, see the date in the Certificate), it means that the reviewers have read the corrected version. Perhaps the comments re English should be addressed to AJE

Regardless of what was said above, we have retyped some places in the text to avoid possible misunderstandings and not fully unambigous statements.

In my opinion, optimization has not been performed in the work - as the title suggests. No models describing the change of the properties of hot-melt adhesives due to variable components of the mixture have been determined.

Thank you very much for the comment, we fully accept you comment for inadequate and misleading term in the title, the title was changed, instead of „optimization“, “modification“  was used.

In my opinion, optimization has not been performed in the work - as the title suggests. No models describing the change of the properties of hot-melt adhesives due to variable components of the mixture have been determined.Optimization is a method of determining the best (optimal) solution (searching for the extreme of a function) from the point of view of a specific criterion (indicator). Testing whether a specific additive reduces or increases a certain property of a material is not an optimization. Additionally, the study did not investigate the interactions related to the simultaneous presence of several components in the mixture. The authors of the paper often define the changes as significant, although the paper does not present even the simplest statistical analysis determining the significance of the observed changes. After such an analysis, it might turn out that some modifications do not affect the properties of the obtained glued joints.

The reviewer is completely right, we appreciate the comment very much, the title of the MS was changed.

The research methodology should be clarified. The method of activating the polymer with O3 (time, temperature, polymer form, apparatus) has not been sufficiently described. The process of preparing the mixture should be more explained. For example: at what point and what ingredients were added to the mixer. The description shows that only Licocene PP 2602 was mixed with the monomer - at which point the remaining ingredients were added? How was Escorez 5600 or Licowax introduced? How was the sample preheated to 80 °C when the temperature of 120 °C was left on the chamber?

Two different procedures were used, first one is a procedure where Licocene was grafted by AA, this was done separately using just ozonized LC and monomer. The second procedure was used for preparation of HMA, where only small amount of grafted LC was added, see the Table 1. The attitude should be clear since two separate parts are seen in the MS, namely 2.2. Acrylic acid grafting... and 2.3. Experimental HMA preparation...

Table 1 does not include all analyzed systems.

The text below (marked red) was added. The composition of the basic material is shown in Table 1. The variations in composition given in Table 1 for materials HMA prepared to estimate the effect of other components are described in legends of the corresponding Tables or Figures.  

The description of the research results is also difficult to understand. The process of preparing the mixture is imprecise and general. It does not show at what point which component was used - heating 0-5 s? suitably the 5-10 S monomer? - monomer was added gradually. No explanation of the increase in torque and temperature of the mixture after 5 minutes (maybe it is due to my misunderstanding resulting from the lack of diligence in the editing of the text).

The procedures in most papers should serve to give general knowledge on the important features of the procedures. The details of the procedures are given usually only in technological papers and, of course, in patents. Otherwise the details may be misleading since many of them are changing if different equipment is used, even if another operator is working. I am afraid that writing all details the reviewer asks for would need three additional pages of space.

After reading the reviewer’s comment we realized that the text is not partially unundestabdable. We decided to delete the Figure 3 and to move the relevant explanation to Experimental part.

There is no analysis of the presented results of rheological tests. Rheological studies are incomplete and do not take into account all the tested systems.

No rheological tests were published in the paper. Would you, pls, let us know where are the rheological tests published in the MS?

There are really many such uncertainties in the manuscript and I will not mention them anymore.

No comment, no response

The list of literature has been made carelessly and there are no recent reports.

Few references from 2020 and 2021 were included. It is not clear what the reviewer means with „carelessly made list of literature“, therefore no response to this part of the comment.

In my opinion, the article should be thoroughly revised. In my opinion, the scientific level of the presented research is inadequate to the rank of the Polymers journal

No comment at all.

Reviewer 2 Report

Dear Authors,

The manusript is overall well writen. I have some minor remarks and comments, which can be found in the attached document.

Best regards

Author Response

REVIEWER 2

All comments were included directly in the copy of the MS. This is useful and time-saving way for the authors to deal with the comments knowing exactly what the reviewer has on his/her mind.

We accepted all comments, pls, check the changes marked red directly in the text of the MS.

 Conclusions were rewritten as well, the comment of long last sentence is appreciated very much, I hope we suceeded to improve it to acceptable level.

Few quotations in the introduction were also inserted related to sentences appointed by the reviewer.

Fig. 1 was deleted, re the Fig 2, (now Fig 1) more details were inserted in the Legend, hopefully it will be more understandable.

Reviewer 3 Report

The authors present grafting of acrylic acid onto an oxygen/ozone-activated metallocene ethylene-co-polypropylene copolymer. The grafted copolymer is applied as a component in a metallocene polyolefin based hot-melt adhesive composition with increased adhesion. The obtained results could be suitable for the practical application. The investigation will be interesting in polymer field and the paper could be published after revision.

- Authors of the manuscript should explain why grafting of acrylic acid was chosen for the grafting ?

- Acrylic acid (AA) (99% monomer purity, Aldrich) was stabilized with 180-200 ppm 4-methoxyphenol.  Was the stabilizer removed before grafting ?

- The optimized composition of the reaction mixture consisted of basic LC polymer mixed with 18.5 wt % of acrylic acid, while 0.5 wt % of NaLS wetting agent was added. How and why the wetting agent was chosen ?

- Composition of the reaction mixture consisted of basic LC polymer mixed with 18.5 wt % of acrylic acid. After completing the grafting reaction, the content of acrylic acid in the final product was 11.4. wt %. Was the unreacted content of the acrylic acid removed from the final product ?

-Figures 1 and 2 do not give a scientific information for the paper ?

-Chemical structures of the used materials and grafting mechanism should be demonstrated in the paper.

-- The authors should describe advantages and disadvantages of the presented hot-melt adhesives as compared with those described in literature ?

-Properties of the developed hot-melt adhesives should be compared with commercially available products ?

Author Response

-The authors present grafting of acrylic acid onto an oxygen/ozone-activated metallocene ethylene-co-polypropylene copolymer. The grafted copolymer is applied as a component in a metallocene polyolefin based hot-melt adhesive composition with increased adhesion. The obtained results could be suitable for the practical application. The investigation will be interesting in polymer field and the paper could be published after revision.

- Authors of the manuscript should explain why grafting of acrylic acid was chosen for the grafting ?

 - Acrylic acid (AA) (99% monomer purity, Aldrich) was stabilized with 180-200 ppm 4-methoxyphenol.  Was the stabilizer removed before grafting ?

The responses for both comments have been merged.

Besides the fact that it is a polar molecule, acrylic acid is a very reactive monomer in polyaddition reaction initiated by free radicals. This is also the reason why 4-methoxyphenol was not removed from the monomer during grafting as it controls the length of grafted chains. The latter should be moderate to bring a required effect.

No related comment was inserted in the MS. I considered to add this in the MS but it would broaden the topic and would need additional discussion over the intended focus to HMA.

- The optimized composition of the reaction mixture consisted of basic LC polymer mixed with 18.5 wt % of acrylic acid, while 0.5 wt % of NaLS wetting agent was added. How and why the wetting agent was chosen ?

The role of the wetting agent is to  increase the homogenity of the reaction system. We are working a lot with wetting agents, NaLS is one of two or three most frequently applied, lot of experience exists on this topic in the Institute. One of the three is in most cases the first choice unless there is a literature recommendation for another wetting agent.  

- Composition of the reaction mixture consisted of basic LC polymer mixed with 18.5 wt % of acrylic acid. After completing the grafting reaction, the content of acrylic acid in the final product was 11.4. wt %. Was the unreacted content of the acrylic acid removed from the final product ?

In our view, acrylic acid reacted completely. 18.5 % is related to the initial system mass while 11.4 % is related to the weight of grafted polymer. It does not mean that these numbers shoud be identical. Moreover, part of the AA might undergo homopolymerization. This was neither analyzed nor removed. If maximum 0,5 wt % of the grafted LC was added to the HMA, when forming the HMA, the amount of polyacrylic acid present is negligible. The polyAA should be almost inert in the system of HMA,

-Figures 1 and 2 do not give a scientific information for the paper ?

Since the method used for evaluation of the adhesive strength towards paper is the IP of Slovak company VIPO and it was published only as the Slovak patent, generally unavailable for readers, we tried to let the readers to know at least the principle of the procedure. Anyway, the Fig 1 was deleted but the legend to the Fig 1 (now being Fig 1) was extended a little,

-Chemical structures of the used materials and grafting mechanism should be demonstrated in the paper.

The main reaction leading to grafting was inserted. The text below was inserted in the MS

Hydroperoxidation of tertiary carbons in propylene moieties by ozone in the presence of oxygen involves the depletion of ozone to molecular oxygen (may be seen on the right hand side of the above  equation) and oxygen atoms while oxygen is subsequently inserted into C-H bond.

The decomposition of hydroperoxides into free radicals  in the next step would initiate their addition to acrylic acid  and chains of polyacrylic acid remain grafted on metallocene substrate.

The chemical composition of all used materials should be clear from the names of the species. Exact chemical names of all chemicals are given in the experimental part.

-- The authors should describe advantages and disadvantages of the presented hot-melt adhesives as compared with those described in literature ?

We believe that the use of activation of hydroperoxides on metallocene copolymers followed by consequent graftinig may substantially reduce the probability of essential changes in key parameters rand kay ultimate properties of the modified HMA. Even rather low concentration of the grafted polymer - based additive leads to significant increase in the adhesive strength between the HMA and paper.

-- The authors should describe advantages and disadvantages of the presented hot-melt adhesives as compared with those described in literature ?

We found only few references on HMA applied for binding the books’ backs. The described HMA exhibits higher adhesion strength, but the melt stability is lower, compared to e.g. Planamelt R produced by Planatol GmbH, Germany, used as the reference material with tensile strength of the adhesive joint between HMA and paper 81 N/90mm. The other parameters – viscosity, solidification time, open time are similar for both HMAs.

-Properties of the developed hot-melt adhesives should be compared with commercially available products ?

The commercial products are the Planamelt R mentioned in the previous response and hopefully the new HMA from VIPO production planned to be introduced in Slovak market within 6 – 9 months.

Reviewer 4 Report

The revised version is fully acceptable for publication.

Author Response

No changes have been suggested by the reviewer. We are appreciating very much the positive opinion of the reviewer

Reviewer 5 Report

This revised paper describes optimization of hot melt adhesives using metallocenes. This is an interesting paper but there are some issues to address before acceptance.

On p. 2, the reaction as written makes no sense. The product O + O2 will recombine to form O3. There are not going to be free O atoms roaming around under their conditions. This is not correct chemically and needs to be fixed.

On p. 3 and 4 in section 2.2, most of the added text in red is redundant and repetitive. Please fix this and do not repeat text. Put the data in a table to shorten this section and make it not repetitive.

The authors have done a very nice job of including error bars. They need to propagate the error bars into values that they are calculating using these experimental values.

The biggest issue with the paper is the redundancy using the same text in section 2.2.

Author Response

COMMENT This revised paper describes optimization of hot melt adhesives using metallocenes. This is an interesting paper but there are some issues to address before acceptance.

 On p. 2, the reaction as written makes no sense. The product O + O2 will recombine to form O3. There are not going to be free O atoms roaming around under their conditions. This is not correct chemically and needs to be fixed.

RESPONSE I apologize very much for rather detailed response to this comment.

In the first version of the MS no reaction was inserted, the reaction as appeared in the rewritten version was inserted due to a request of a reviewer. This response was prepared by Dr Jozef Rychly who is dealing with oxidation and antioxidant matters as well as incinneration and flame retardancy of plastics for more than 50 years. Besides being considered as the most acknowledged expert in the field in Slovakia, he is also very modest person and nice to all people. This was the reason that, unfortunately, instead of refusing to add the reaction scheme and explaining the reason, he just selected one of possible (published) schemes which seemed to be the simplest one. After your comment he offered following response.

RESPONSE OF DR RYCHLY I am sorry but various schemes of interaction of ozone with metallocene polymers may be put down and it is true that the approach in the MS is not optimal.

From several  alternative reaction schemes  the most likely seems that ozone interaction with C=C unsaturations present in metallocene polymer leading to Criegee molozonides with  subsequent cleavage and formation of e.g. carbonyl hydroperoxides as shown in paper REFERENCE INSERTED.

THE SCHEME IS INSERTED IN THE SECOND REWRITTEN VERSION OF THE MS.

The further scission of these hydroperoxides leads to grafting eventually to the reduction of polymer chain length - REFERENCE.

Regarding zwitterionic character of ozone molecules there may other alternative mechanisms, however, the Criegee alternative seems to be the fastest.

CONTINUING MY RESPONSE However, since several alternatives are mentioned, I would prefer not to stick up with any of these, the exact scheme is not of key importance considering the topic of the paper which is, by the way, of rather applied nature. Formation of peroxides and hydroperoxides able to initiate grafting was clearly demonstrated, so I would prefer to omit the scheme and to explain the reason for the reviewer.

Anyway, after a discussion with Dr Rychly, I accepted to insert his suggestion, slightly modified by myself, which is included in the MS supported by two new references. If in your view the omitting the exact scheme is more appropriate, we are ready to make this change asap.

We appreciate very much your time and expecience in reviewing our MS.

COMMENT On p. 3 and 4 in section 2.2, most of the added text in red is redundant and repetitive. Please fix this and do not repeat text. Put the data in a table to shorten this section and make it not repetitive.

RESPONSE The text was deleted and carefully checked.

I apologize very much for this inconsistency. It happened since the deadline for changes was set to too short period, certain parts of the manuscript were modified by several authors. It happened that two colleagues took care of the same part of the MS and both inserted the same part to the final version. Of course, I was responsible for the final version of the MS, but I was submitting the paper by late night on the last day of the deadline indicated by the editor and my attention was too low to read whole text carefully enough. In fact I was sure that in experimental part nothing inappropriate can be found, so simply wnt through it briefly and I did concentrate on the text sufficiently.   

COMMENT The authors have done a very nice job of including error bars. They need to propagate the error bars into values that they are calculating using these experimental values.

RESPONSE I would prefer to leave the figures at current shape, since presenting the results in Tables seems not  nice, the Tables consist of one line each. Tentatively I prepared the Figures based on column graphs with all numerical data inserted in the picture.

COMMENT The biggest issue with the paper is the redundancy using the same text in section 2.2.

RESPONSE Done

Round 2

Reviewer 1 Report

First, I would like to refer to the authors' responses to my previous commentary. The authors are responsible for the manuscript and I will have no claim to the institution that made the linguistic proofreading. If the English language is correct, it means that the text is written incomprehensibly.

If the article requires it and you need to write three additional pages, do it. Authors should make every effort and spend enough time that the reader does not have to tired reading the article. Please respect the time of the reviewers. The task of the reviewers is to evaluate the scientific content of the article, and not to complete editorial and stylistic proofreading.

The manuscript by the authors has not been read with understanding and has been properly revised. It still contains a lot of errors.

I am just giving two examples: Plastographometric tests - instead of explaining, the plastogram was removed. It's a pity. The text has been moved to the methodology in which there are now repetitions. Please pay attention to the following sentence: After filling the chamber with the polymer and closing the chamber, temperature increased to about 80 °C within 5 to 10 seconds…..

I have performed thousands of plastographometric tests and I have never experienced such a rapid heating of the material. It was necessary to carefully analyze the plastogram. I guess it was about minutes, not seconds?

The authors claim “No rheological tests were published in the paper”. Is viscosity measurement not rheological tests? Table 1 - Brookfield viscosity 170 ° C. Table 1 (as well as the entire manuscript) contains editorial errors - which indicates insufficient diligence of the authors in the preparation of the manuscript.

I maintain my opinion that the level of the presented manuscript does not permit its publication in Polymers.

Author Response

COMMENT First, I would like to refer to the authors' responses to my previous commentary. The authors are responsible for the manuscript and I will have no claim to the institution that made the linguistic proofreading. If the English language is correct, it means that the text is written incomprehensibly.

RESPONSE I just wonder whether you ever cooperatged with a professional agency providing corrections of scientific papers. The top agency, such as AJE, cooperates with native English speaking people, which are at the same time experienced scientists working in the area closely related to the submitted manuscript. They do not only correct the English but also suggest changes in sentences to make the result fully understandable. If I get the corrected text, almost always there are few places with a note THE SENTENCE WAS AMENDED, PLS, CHECK WHETHER THE MEANING WAS NOT CHANGED. You may be sure that the person who made the corrections, understood each detail of the manuscript. Therefore, it is impossible that my text could be written incomprehensibly for a reader who understands Engllish reasonably.

COMMENT  If the article requires it and you need to write three additional pages, do it. Authors should make papers every effort and spend enough time that the reader does not have to tired reading the article. Please respect the time of the reviewers. The task of the reviewers is to evaluate the scientific content of the article, and not to complete editorial and stylistic proofreading.

RESPONBSE I am definitely sure that this article does not need to be extended by three additional pages. in my view, as well as according to the opinion of the other four reviewers, the manuscript needs few changes, all of them have been well defined by all the reviewers except for you. Further on, I do not need to be advised what is the task of the reviewers, I am reviewing papers and projects for more than 30 years, during last 15years around 20 to 40 papers a year.

COMMENT / RESPONSESS IN BOLD The manuscript by the authors has not been read with understanding  I DO NOT QUITE UNDERSTAND WHAT YOU ARE MEANING  BY THIS STATEMENT and has been properly revised. IN THE SECOND PART OF YOUR SENTENCE YOU SURPRISINGLY CLAIM THAT THE REVISION OF THE MANUSCRIPT WAS CORECT AND SUFFICIENT. It still contains a lot of errors.

COMMENT I am just giving two examples: Plastographometric tests - instead of explaining, the plastogram was removed. It's a pity. The text has been moved to the methodology in which there are now repetitions. Please pay attention to the following sentence: After filling the chamber with the polymer and closing the chamber, temperature increased to about 80 °C within 5 to 10 seconds…..

RESPONSE You are right, after removing the Figure and transfering a part of the text to experimental section, some repetition occured, as also another reviewer mentioned. Thank you for this note, the respective section was carefully read and corrected. I hope you will accept this change.

Thank you very much for mentioning the rapid heating of the material in the chamber. In fact the real situation was like this: The temperature in the chamber was set to 120 oC, the thermometer in the chamber was attached very close to the side wall of the chamber (standard construction of that type of the Brabender chamber), so for empty chamber the temperature of 120 oC was registered. After opening the chamber, temperature went down and during filling the chamber with polymer powder kept at room temperature, the temperature DECREASED  to about 80 oC. The person performing the mixing and having written the corresponding experimental part made a mistake writing „temperature increased“. In the corresponding Figure, it is seen that at the beginning, temperature is really decreasing. Obviously, if the chamber was set to 120 oC, after touching the powder taken at RT from the shelf, the thermometer was colled down.

In any case, I prefer not to insert the Figure, it has no importance concerning the topic of the paper, which is aimed to the ultimate properties of the hot melt adhesive after modification and not to details of grafting.

COMMENT I have performed thousands of plastographometric tests and I have never experienced such a rapid heating of the material. It was necessary to carefully analyze the plastogram. I guess it was about minutes, not seconds?

RESPONSE The time was correctly claimed in second, the crucial mistake consisted in the statekemt that temperature was increased. By the way I fired the guy making sich mistakes.

COMMENT The authors claim “No rheological tests were published in the paper”. Is viscosity measurement not rheological tests? Table 1 - Brookfield viscosity 170 ° C. Table 1 (as well as the entire manuscript) contains editorial errors - which indicates insufficient diligence of the authors in the preparation of the manuscript.

RESPONSE Brookfield vioscosity measurements were performed for the final HMA composition to confirm the effect of the additive on the practical performance of the composition. No „study“ (as you required in your previous review) was made, the viscosity data were similar without showing any tendency, this conclusion was expected, so in my view, no discussion is needed.

COMMENT I maintain my opinion that the level of the presented manuscript does not permit its publication in Polymers

RESPONSE I am respecting your decision, however, the other four reviewers seem to have opposite opinion.

Reviewer 3 Report

If editor and other reviewers agree that the paper is suitable for “Polymers” I recommend the paper for publication after the revision.

Author Response

No additional changes have been suggested by the reviewer. We are appreciating very much the positive opinion of the reviewer